# Game-theoretic Counterfactual Explanation for Graph Neural Networks

## ABSTRACT

Graph Neural Networks (GNNs) have been a powerful tool for node classification tasks in complex networks. However, their decision-making processes remain a black-box to users, making it challenging to understand the reasoning behind their predictions. Counterfactual explanations (CFE) have shown promise in enhancing the interpretability of machine learning models. Prior approaches to compute CFE for GNNS often are learning-based approaches that require training additional graphs. In this paper, we propose a semivalue-based, non-learning approach to generate CFE for node classification tasks, eliminating the need for any additional training. Our results reveals that computing Banzhaf values requires lower sample complexity in identifying the counterfactual explanations compared to other popular methods such as computing Shapley values. Our empirical evidence indicates computing Banzhaf values can achieve up to a fourfold speed up compared to Shapley values. We also design a thresholding method for computing Banzhaf values and show theoretical and empirical results on its robustness in noisy environments, making it superior to Shapley values. Furthermore, the thresholded Banzhaf values are shown to enhance efficiency without compromising the quality (i.e., fidelity) in the explanations in three popular graph datasets.

## KEYWORDS

Explainable AI, Counterfactual Explanation, Graph Neural Network

**ACM Reference Format:**
Anonymous Author(s). 2018. Game-theoretic Counterfactual Explanation for Graph Neural Networks. In *Proceedings of Make sure to enter the correct conference title from your rights confirmation emai (Conference acronym 'XX).* ACM, New York, NY, USA, 12 pages. https://doi.org/XXXXXXX.XXXXXXX

## 1 INTRODUCTION

Graph Neural Networks (GNNs) have recieved substantial attention in recent years due to their remarkable success in diverse domains, including computational biology and drug discovery [10, 11], natural language processing [40, 46], and computer security [44, 48]. However, GNNs are seen as black-box models, making model explainability a crucial aspect for their practical deployment. Counterfactual explanations (CFE) have emerged as a significant approach for elucidating the predictions of GNN models and providing algorithmic recourse [14]. CFE's primary objective is to find the minimum input modifications required to change the model's prediction,

*Conference acronym 'XX, June 03–05, 2018, Woodstock, NY*
© 2018 Association for Computing Machinery.
ACM ISBN 978-1-4503-XXXX-X/18/06...$15.00
https://doi.org/XXXXXXX.XXXXXXX

offering valuable insight in applications where human oversight is indispensable[47].

Existing learning-based approaches for CFE often necessitate the training of additional graphs, which can be computationally intensive and the procedures themselves might lack interpretablity. For instance, CF-GNNExplainer [25] requires a model training with the negative log-likelihood loss. In this work, we concentrate on designing a *non-learning-based approach* grounded in semivalues [8], such as Shapley and Banzhaf values. Semivalues have seen widespread application in explaining machine learning models in various domains and have also found utility in similar fields like data valuation [20, 22, 39]. In essence, semivalues can provide effective solutions to problems that can be formulated as cooperative games. Given that the problem of CFE generation for node classification (as shown in Fig. 1) can be formulated into this category, we propose a method for utilizing semivalues to address this problem.

Most prior work in the area of explainable AI (XAI) that applies semivalues formulates the problem of attribution as cooperative game theory where the utility $U$ needs to be divided among some players $N$. These can be nodes of a graph or tree, features of models, etc. Most of them, with a few notable exceptions [6, 17, 49], use Shapley values [36] for this attribution. However, more recently it has been shown that Shapley values may not be best choice because of their lack of mathematical and human-centric intuition suitable for attributing explanations [19]. Therefore, in this work, we focus on another popular semivalue called Banzhaf values [4] and show that they have desirable properties such as robustness in the presence of noise and computational efficiency. We provide both theoretical and empirical evidence supporting the idea that our method using Banzhaf values offers robustness and computational efficiency, which is also consistent with prior work [39].

We introduce the idea of using semivalues in combination with thresholded utility functions. We provide intuition for how adding a threshold to the utility functions can make semivalues more reliable. We also examine the concept of a safety margin from Wang and Jia [39], which represents the minimum amount of noise needed in the worst case to change the influence of any two edges in a node's edge set. We show that thresholding does not alter the safety margin for any semivalue due to our use of a hinge function to perform the thresholding. Since Wang and Jia [39] show that Banzhaf values achieve a higher safety margin than Shapley, it follows that thresholded Banzhaf values are also superior to thresholded Shapley. Our experimental results show that, in addition to adding robustness, using thresholding can reduce the computation time for Banzhaf values.

Our contributions can be summarized as follows:

- We propose a semivalues-based, non-learning based method for generating CFE for node classification task. Our method does not require additional training of a GNN model.
- We show that Banzhaf values have a lower sample complexity of finding the best $k$-explanations than Shapley values. Our

**Figure 1: Figure contains two graphs. Each nodes in the graph belong to either the class pink (nodes that are present in the 5-node cycle motif) or green. Deleting the edge in the graph (right) leads to flip in the classification of node B (pink becomes green). Here, the deleted edge is the counterfactual explanation for node B.**

experimental results also validate that our method based on computing Banzhaf values is up to 4 times faster in finding the best $k$ explanations than computing Shapley values.

- We demonstrate both theoretically and empirically that our thresholding method for computing Banzhaf values is more robust than Shapley values in the presence of noise. Our theoretical analysis on thresholding can be applied to any semivalue. However, we prefer Banzhaf values because of its efficiency, robustness and intuitive power.

- We empirically show that, apart from robustness, adding the thresholded Banzhaf values can gain in efficiency over non-thresholded Banzhaf values computing the CFEs without compromising on fidelity values in most cases. In many cases with noisy GNN classifiers, the fidelity is also improved.

## 2 PRELIMINARIES AND NOTATION

### 2.1 Graph neural networks

Consider a graph, denoted as $G = (V, A)$, consisting of a set of nodes ($V$) and a set of edges ($E$). Let $X \in \mathbb{R}^{n \times d}$ represent the $d$-dimensional features of $n$ nodes in $V$, while $A \in \{0, 1\}^{n \times n}$ is the adjacency matrix specifying edges in the edge set $E$. Graph Neural Networks (GNNs) [13, 18, 38] have proven to be effective in making predictions on such graphs by learning relevant low-dimensional node representations through a message-passing mechanism.

During message passing, each node ($u \in V$) updates its representation by aggregating information from itself and its set of neighbors $N(u)$. Mathematically, the update in $l$-th step can be represented as follows:

$$h_u^{(l)} = AGGR(h_u^{(l-1)}, \{h_i^{(l-1)} | i \in N(u)\}) \quad (1)$$

where $h_u^{(l)}$ is the updated representation of node $u$ at iteration $l$, obtained by applying the aggregation operation ($AGGR$) to combine its previous representation ($h_u^{(l-1)}$) with those of its neighboring nodes. The representation at the 0-th step is the initial feature set of the nodes. GNNs iteratively apply this equation to refine the node representations, capturing the structural patterns and dependencies within the graph for a wide range of tasks such as node classification, link prediction, and graph. classification. For a more detailed introduction and applications of GNN we refer the reader to the survey [51].

## 2.2 General Definitions of Semivalues

In this work, we utilize cooperative game theoretic techniques to build a counterfactual explainer for GNNs. More specifically, we utilize semivalues to compute the impact of each edge that are edited in generating the counterfactual explanation. While existing counterfactual explainers often require additional training, computing semivalues can circumvent this need, making it a preferable choice in practice. Here, we introduce the general definition of semivalues and two popular approaches, namely Banzhaf [4] and Shapley values [36].

*2.2.1 General Semivalues.* Cooperative game theory offers a mathematical framework for analyzing the distribution of gains or contributions among a coalition of agents or players while satisfying some desired properties in the resulting distribution. For example, the Shapley value (denoted by $\phi$ here) satisfies the following properties [12, 15]:

(1) Linearity: $\phi(v_1 + v_2) = \phi(v_1) + \phi(v_2)$
(2) Dummy player: If $U(S \cup i) = U(S) + c$ for all $S \subseteq N \setminus \{i\}$ and some $c \in \mathbb{R}$, then $\phi_i(U) = c$.
(3) Symmetry: If $v(C \cup \{i\}) = v(C \cup \{j\})$ for all $C \subseteq N \setminus \{i, j\}$, then $\phi_i(v) = \phi_j(v)$.
(4) Efficiency: for every $U, \sum_{i \in N} \phi_i(U) = U(N)$

Mathematically, semivalues satisfy all these properties except the *efficiency* property. A semivalue is defined as a function $\phi_{semi}$ that maps a coalition $S \subseteq N$, where $N$ is the set of all agents, to a real number $\phi_{semi}(S)$, representing the value or contribution of the coalition $S$. Due to their appealing properties, many variants of semivalues have been proposed in the literature [5, 12, 17, 20, 21, 31, 37, 45]. We state the following result which gives a useful characterization of the form of semivalues.

THEOREM 1. *(Dubey and Shapley [8]). A value function $\phi_{semi}$ is a semivalue, if and only if, there exists a weight function $w : [n] \rightarrow \mathbb{R}$ such that $\sum_{j=1}^{n} \binom{n-1}{j-1} w(j) = n$ and the value function $\phi_{semi}$ can be expressed as follows:*

$$\phi_{semi}(i; U, w) := \sum_{j=1}^{n} \frac{w(j)}{n} \sum_{S \subseteq N \setminus \{i\}, |\bar{S}| = j-1} [U(S \cup i) - U(S)] \quad (2)$$

*2.2.2 Shapley Value.* The Shapley value [36] is a well-known semivalue that has been popular to compute the importance of the features in machine learning models[17]. It provides a fair way of distributing the value generated by a coalition among its members. The Shapley value of an agent $i$ is calculated as:

$$\phi_i = \sum_{S \subseteq N \setminus \{i\}} \frac{(|S|!(|N| - |S| - 1)!)}{(|N| - 1)!} (U(S \cup \{i\}) - U(S)) \quad (3)$$

where $|S|$ represents the size of the coalition $S$, $|N|$ is the total number of agents, $U(S)$ represents the worth of coalition $S$, and $U(S \cup \{i\})$ represents the worth of coalition $S$ with the agent $i$ when $i \notin S$. The Shapley value $\phi_i$ quantifies the average contribution of the agent $i$ in all possible coalitions involving $i$.

*2.2.3 Banzhaf Index.* The Banzhaf index ($\beta$) [4] is another popular semivalue in cooperative game theory. It measures the influence or

power of an agent in a cooperative game. The Banzhaf index ($\beta_i$) of agent $i$ is computed as:

$$\beta_i = \frac{1}{2^{(|N|-1)}} \sum_{S \subseteq N \setminus \{i\}} (U(S \cup \{i\}) - U(S))$$

where the terms have the same meanings as in Eq. 3. Intuitively, $\beta_i$ quantifies marginal value of agent $i$'s presence, considering all possible coalitions.

In this work, we employ both the Shapley and Banzhaf values to assign contributions to individual edges regarding the classification decisions made by GNNs while leveraging their properties. Nevertheless, our theoretical and empirical analyses (Sec. 4 and 5) suggest that the Banzhaf values offer superior performance in terms of interpretability, robustness, and complexity.

## 3 PROBLEM DEFINITION

Our objective is to address the problem of generating concise counterfactual explanations for the predictions given by graph neural networks (GNNs) on node classification tasks. In a given graph $G = (V, E)$, each node $v \in V$ is associated with a feature vector $x_v \in \mathbb{R}^d$. Furthermore, $L(v) : v \to C$ is a function that maps each node $v$ to its true class label drawn from a set $C$. Now there exists a classifier GNN $\Phi$ that has been trained on $G$. For the node classification task, given an input node $v \in V$, we assume $\Phi(G, v, c)$ outputs a probability distribution over class labels $c \in C$. The predicted class label is therefore the class with the highest probability, which we denote as $L_\Phi(G, v) = \arg\max_{c \in C} \{\Phi(G, v, c)\}$.

The idea is to delete a few edges from the graph such that the predictions of the GNN for the node $v$ changes to a different one from the original prediction. The set of deleted edges becomes the counterfactual instance for classifying that particular node $v$.

**PROBLEM STATEMENT** 1 (COUNTERFACTUAL EXPLANATION FOR NODE CLASSIFICATION). *Given an input graph $G = (V, E)$, a budget $k$, a node $v$, a GNN model $\Phi$, we aim to identify a set of edges $S^*$ (where $|S^*| = k$ and $S^* \subset E$) to be deleted such that $L_\Phi(G^*, v) \neq L_\Phi(G, v)$ and $G^* = (V, E \setminus S^*)$.*

We aim to design an algorithm to produce a solution set of edges that act as a counterfactual explanation. This set of edges will be evaluated based on the well-known measures for explanations such as *Fidelity* [25] which is the proportion of nodes whose predicted class remains the same after the explanation set of edges is deleted.

**Importance functions from cooperative games.** Problem 1 asks for a set of edges as a counterfactual explanation. A common method to formalize the explanations in machine learning models involves feature importance function [26]. In our problem context, "feature" refers to the edges within the given graph. Thus, we change our original objective in Problem 1 and *simplify* our objective in selecting $k$ edges such that the importance of the set $S^*$ is maximized. We can write our objective as follows:

$$S^* = \arg \max_{S \subset E, |S|=k} \sum_{e \in S} IMP(e)$$

where $IMP$ is the importance function. However, coming up with a good importance function is difficult as the interactions between the edges needs to be captured in it. For example, while both edges

$e_1$ and $e_2$ might be important but they also have to be complimentary, so that their combined effect is visible in the task when they are deleted simultaneously. Note that this simplified objective is effective in producing counterfactuals in practice (Sec. 5.2).

*We design the importance function from the utility functions in cooperative game theory.* The edges in $E$ represent all the players in the corresponding game. Given the node $v$ and its initial predicted class by $\Phi$ is $c$; the utility function for any set of edges $S$ is the reduction in the initial prediction probability when $S$ is deleted from the graph. More formally, the utility function is $U : 2^E \to R$ takes any subset of edges $S \in E$ and maps it to the decrease in probability of the current assigned class:

$$U(S) = \Phi(G, v, c) - \Phi(G', v, c), \text{ where } G' = (V, E \setminus S) \qquad (4)$$

Our goal is to find the set of edges $S^*$ (where $|S^*| \leq k$) such that $U(S^*)$ is maximized. While prior work has computed the similar objective for feature importance using Shapley values [26], we theoretically and empirically show that for CFE, Banzhaf values produce results with higher quality and robustness (Secs. 4 and 5).

## 4 OUR METHOD: THRESHOLDED BANZHAF VALUES

In this section, we design our algorithm based on Banzhaf values and describe its advantages over using Shapley values. The theoretical results suggest that Banzhaf values have several advantages over Shapley values such as computational efficiency and robustness. Formally, Banzhaf value of player $i$ in cooperative game is the average of all marginal utilities $\Delta_i = U(S \cup i) - U(S)$ of player $i$. For counterfactual explanations, the $U(S)$ is the decrease in classification probability for the current (undesired) class caused by deletion of a set or coalition of edges.

Besides the above advantages, Banzhaf values are more intuitive than Shapley values in our problem context. Banzhaf values take an expectation over the marginal contribution in all the coalitions whereas Shapley values take an expectation over the average contribution in all possible orders the coalition could have formed. That is, Banzhaf values can be more intuitive for counterfactual explanation because they can be directly seen as *expected drop in the probability caused by an edge $i$ in various coalitions*. On the other hand, Shapley values are hard to interpret because it considers all possible ordering of coalitions and adding different edges.

In the subsequent sections, we discuss the advantages of Banzhaf values. In Sec. 4.1, we discuss the advantage of computational efficiency and explain why the Banzhaf value computation has lower time complexity. Next, we discuss that Banzhaf values can be made even more efficient and scalable by adding thresholding to the utility function (Sec 4.2). We also validate emperically that Banzhaf values with thresholding lead to a significant improvement in computational efficiency. Finally, we prove that that Banzhaf values with a threshold keep the same robustness properties of vanilla Banzhaf values (Sec 4.3).

### 4.1 Computational Efficiency

In this subsection, we demonstrate that one of the primary advantages of Banzhaf values over Shapley values lies in their computational efficiency. Specifically, sicne the CFE problem necessitates

identifying the top-$k$ edges as the explanation set, we illustrate that determining the top-$k$ Banzhaf values involves a complexity that is lower by a factor of $O(n)$ compared to the computation of Shapley values. To this end, we first outline the methodology for obtaining the Banzhaf values and subsequently provide sample complexity required to derive the final solution set.

Much like the Shapley value and other data value concepts based on semivalues, the precise computation of the Banzhaf value can be prohibitively expensive due to its dependence on an exponential number of utility function evaluations and the need to eliminate an exponentially large number of coalition combinations. To address this issue, we introduce the Monte Carlo estimator $\phi_{MC}$ for estimating both the Banzhaf and Shapley values. Furthermore, we leverage the Maximum Sample Reuse (MSR) estimator $\phi_{MSR}$ for the Banzhaf values as proposed by [39], which results in improved time complexity for Banzhaf calculations. .

**Monte Carlo Estimator**. For estimating Banzhaf and Shapley, the Monte Carlo approach is a standard approach. Since the computation is similar for Shapley values, we just briefly describe the Banzhaf value computation as per equation (5). The Banzhaf value for an edge $i$, can be approximated by randomly sampling data subsets from the power set of edges $2^N$, excluding the edges of interest $i$ as $\widehat{\phi}_{MC}(i) = \frac{1}{|S_i|} \sum_{S \in S_i} [U(S \cup \{i\}) - U(S)]$. This straightforward Monte Carlo method involves sampling subsets uniformly and then computing the contributions of these subsets to the value. The resulting approximated semivalue vector, denoted as $\widehat{\phi}_{MC}$, consists of the estimated Banzhaf values for each edge $i$.

$$\phi_{\text{Banzhaf}}(i) = \mathbb{E}[S \sim \text{Unif}(2^{N \setminus \{i\}})] [U(S \cup \{i\}) - U(S)] \quad (5)$$

**MSR Estimator**. The MSR estimator addresses the sub-optimality of the simple Monte Carlo (MC) method for estimating Banzhaf values. In the simple MC method, each sample value of $U(S)$ and $U(S \cup i)$ is used only for estimating the Banzhaf values of edge $i$; where as in MSR estimate, the $U(S)$ and $U(S \cup i)$ are used to estimate for all $i \in S$. Using linearity of expectation, we can write the Banzhaf value $\phi_{\text{Banzhaf}}(i)$ for the edge $i$ as:

$$\phi_{\text{Banzhaf}}(i) = \mathbb{E}[U(S \cup i)] - \mathbb{E}[U(S)]$$

The MSR estimator, denoted as $\phi_{\text{MSR}}(i)$, estimates $\phi_{\text{Banzhaf}}(i)$ for the edge $i$ with:

$$\phi_{\text{MSR}}(i) = \frac{1}{|S_{\in i}|} \sum_{S \in S_{\in i}} U(S) - \frac{1}{|S_{\notin i}|} \sum_{S \in S_{\notin i}} U(S)$$

where $S_{\ni i} = \{S \in S : i \in S\}$ and $S_{\not\ni i} = \{S \in S : i \notin S\} = S \setminus (S_{\ni i})$. If either $|S_{\ni i}|$ or $|S_{\not\ni i}|$ is 0, $\phi_{\text{MSR}}(i)$ is set to 0. The MSR estimator maximizes sample reuse and significantly reduces sample complexity compared to the simple MC method, making computation of Banzhaf values computationally efficient. The algorithm is described by Algorithm 1. A similar estimator for the Shapley values is not effective due to numerical stability issues [39] and so the sample complexity of Shapley value is $n$ (the number of edges) times more than Banzhaf values .

**Sample Complexity.** Next, we show the sample complexity to compute $\phi_{MSR}$ which is used to obtain the Banzhaf values and compare it with $\phi_{MC}$ which is used to compute the Shapley values. We first begin by analyzing the sample complexity for correctly

---

**Algorithm 1** MSR Estimator for Banzhaf Values

**Require:** Set of samples $S = \{S_1, \ldots, S_m\}$, where each $S_i$ is drawn i.i.d. from $Unif(2^N)$
**Ensure:** Banzhaf value estimate $\hat{\phi}_{MSR}(i)$ for each data point $i \in N$
1: **for** $i \in N$ **do**
2:      Divide $S$ into $S_{\ni i} \cup S_{\not\ni i}$ where $S_{\ni i} = \{S \in S : i \in S\}$ and $S_{\not\ni i} = \{S \in S : i \notin S\} = S \setminus S_{\ni i}$
3:      **if** $|S_{\ni i}| > 0$ and $|S_{\not\ni i}| > 0$ **then**
4:         Compute $\hat{\phi}_{MSR}(i) = \frac{1}{|S_{\ni i}|} \sum_{S \in S_{\ni i}} U(S) - \frac{1}{|S_{\not\ni i}|} \sum_{S \in S_{\not\ni i}} U(S)$
5:      **else**
6:         Set $\hat{\phi}_{MSR}(i) = 0$

---

ranking pairs of edges whose Banzhaf values are sufficiently well separated.[1]

LEMMA 1. *Using $\hat{\phi}_{MC}$, with probability at least $1 - \delta$, all edges $i$ and $j$ with $\beta_i > \beta_j + \epsilon$ are correctly ranked after $\frac{4n}{\epsilon^2} \ln \left( \frac{2n}{\delta} \right)$ calls to the $U(.)$ function.*

PROOF. Let $i$ and $j$ be given with $\beta_i > \beta_j + \epsilon$. Then $\hat{\beta}_i > \hat{\beta}_j$, or $\hat{\beta}_i - \beta_i > \hat{\beta}_j - \beta_i$, if $\hat{\beta}_i - \beta_i > \hat{\beta}_j - \beta_j - \epsilon$. For this it suffices that $\hat{\beta}_i - \beta_i > -\epsilon/2$ and $\hat{\beta}_j - \beta_j < \epsilon/2$.

This holds for all such pairs $i$ and $j$ if $|\hat{\beta}_i - \beta_i| < \epsilon/2$ for all $i$. Via a union bound, it suffices that $P(|\hat{\beta}_i - \beta_i| \geq \epsilon/2) \leq \delta/n$.. Via a Hoeffding bound, we need the number of samples $m$ to satisfy $2 exp(-m\epsilon^2/2) \leq \delta/n$, or $m \geq 2/\epsilon^2 ln(2n/\delta)$. Each sample requires 2 calls to the $U(.)$ function and these samples are used to estimate a single edge so we need $n$ times as many to form estimates for all edges. $\square$

This lemma immediately provides for the correct identification of the top-$k$ Banzhaf values with the number of samples based on the gap between the $k$th and $k + 1$st values.

COROLLARY 1. *WLOG Let $\beta_1 > \beta_2 > \ldots > \beta_n$. If $\beta_k > \beta_{k+1} + \epsilon$ then after $\frac{4n}{\epsilon^2} \ln \left( \frac{2n}{\delta} \right)$ calls to the $U(.)$ function the top $k$ edges are correctly identified with probability at least $1 - \delta$ using the $\widehat{\phi}_{MC}$ estimator. That is, for all $i \leq k$ and $j > k$, $\hat{\beta}_i > \hat{\beta}_j$.*

Next we show that, Maximum Sample Reuse estimator $\hat{\phi}_{MSR}$ requires asymptotically less samples by an $O(n)$ factor than $\hat{\phi}_{MC}$ for estimating all comparison, and thus also the top-$K$ values.

LEMMA 2. *Using $\hat{\phi}_{MSR}$, with probability at least $1 - \delta$, all edges $i$ and $j$ with $\beta_i > \beta_j + \epsilon$ are correctly ranked after $\frac{128}{\epsilon^2} \ln \left( \frac{5n}{\delta} \right)$ calls to the $U(.)$ function.*

PROOF. As before we need to find the number of samples $m$ so that $P(|\hat{\beta}_i - \beta_i| \geq \epsilon/2) \leq \delta/n$. From the proof of theorem 4.9 Wang and Jia [39] for the MSR estimate, it suffices to choose $m$ to satisfy $5exp(-m\epsilon^2/128) \leq \delta/n$, or $m \geq 128/\epsilon^2 ln(5n/\delta)$. Each

---

[1]Wang and Jia [39] provide a sample complexity analysis for a different notion of approximation quality which does not immediately imply our desired propery of correctly estimating the top $k$, so for completeness we provide a full analysis.

sample now requires a single call to the $U(.)$ function and is used in all $n$ estimates. $\square$

In conclusion, computing Banzhaf values is practical in terms of number of samples and is faster than computing Shapley values using the MSR estimator since computing the later is more computationally expensive because of the sample complexity dependency's on additional factor of $O(n)$

## 4.2 Thresholding

**In a Practical Context:** Our goal is to generate counterfactual explanations for a graph neural network (GNN) classifier. One of the major challenges is that in GNNs, noise can arise from various sources, including stochastic or noisy edges, targeted adversarial attacks by adding noise to the data, or inherent variability introduced during classifier retraining via the stochastic gradient descent process. This variability can result in slightly different utility functions for our problem setting, where these utility values are based on the prediction probabilities (see Section 3). Furthermore, in practice, many coalitions of edges may yield low utility values. Sampling such coalitions can lead to high time complexity without providing much information about the contribution of the edges. Additionally, the aggregation of these low-utility coalitions could potentially result in the incorrect computation of the importance of individual edges, ultimately leading to erroneous Banzhaf values.

**Introducing a threshold.** To address these challenges, we introduce a (small) constant threshold in the transformed the utility function. This modification is designed to alleviate the impact of noise and diminish the influence of coalitions with low utility value. Specifically, we transform the utility function in Equation (4) into a hinge function: $U(S) = \max(U(S) - B, 0)$. It is noteworthy that this thresholding approach deviates from the conventional practice in voting games, where utility functions assume values of 0 and 1 below and above a certain threshold value, respectively [1, 52]. Clearly, in the conventional approaches, the utility value after thresholding lacks smoothness. Therefore, slight variations in the threshold value can result in substantially different utility values. As a result, such a utility function will produce Banzhaf values that are highly sensitive to the threshold. Conversely, our thresholding via the hinge function overcomes this issue and fits better in our problem setting. The smoothness of the hinge utility function avoids abrupt changes in measuring the Banzhaf values, ensuring a robust assessment of the importance of the edges in the explanation set.

**Pruning of Coalitions.** Apart from the above advantages, thresholding for Banzhaf values can help in identifying "dummy" coalitions which do not contribute the computation of Banzhaf values. This can especially be true for large coalitions where a single edge might not contribute too much and small coalitions which where adding one or two edges may not contribute any utility. In such cases, we can stop sampling coalitions of particular size or particular edges beyond a certain limit and considerably save the computation time. Our experimental results validates this in multiple datasets (Sec. 5.2). Besides, pruning these coalitions leads to faster computation for Banzhaf values. *Our experimental results demonstrate that the thresholding can yield speed up in running time while maintaining the quality of the explanations (Sec. 5.2).*

## 4.3 Better Robustness

In this section, we discuss the robustness of the thresholding Banzhaf values with the presence of noise. In particular, we prove that adding a reasonable amount of threshold to Banzhaf values will not lead to any changes in the explanations even in the presence of noise. Banzhaf values are known to be to noise-tolerant [39]. We show that Banzhaf values with thresholding also have the same property. Our analysis on thresholding semivalues however is more general and can be applied to any semivalue. Here, we introduce only the key concepts. For detailed definitions of the robustness framework and related background, refer to Appendix A.1.

Our theorem demonstrates that thresholding does not alter the safety margin derived by [39] as it is smooth around the threshold. The safety margin measures the largest amount of noise that can be added to the semivalues (e.g., Banzhaf values) without altering the ranking of any two players (e.g., the edges) $i$ and $j$ in the worst case. A large safety margin indicates that the semivalue is more tolerant to noise. This analysis is pivotal in establishing the consistency and resilience of the safety margin in the presence of thresholded utilities, thereby extending the findings presented in [39] to the case of thresholded semivalues.

THEOREM 2. *Adding the threshold to the utility function doesn't change the safety margin for any semivalue w.*

**Remark**. The proof of Theorem 2 is in Appendix A.1. The proof shows that, in the worst case that defines the safety margin, adding a threshold to the utility function doesn't affect the amount of noise.

Since [39] show that the Banzhaf value achieves the largest safety margin among all semivalues and Theorem 2 shows that the safety margin remains the same in the case of thresholding, we can conclude that Banzhaf values still achieve largest safety margin.

Theorem 2 shows that in the worst case, the threshold doesn't have any effect on the utility function. However, in the general the first inequality in Equation (8) of the proof is strict and adding the threshold helps reduce noise in the utility function. Our experimental results show that in many cases the fidelity improves after thresholding is applied in the noisy GNN.

## 5 EXPERIMENTAL RESULTS

### 5.1 Setup

In this section, we specify the experimental setup for our experiments, including the datasets, the base models, the compared baselines, the evaluation metrics, and the parameters. Our code is in Python and all the experiments have been executed on a Tesla T4 GPU with 16GB RAM.

*5.1.1 Datasets.* We evaluate our algorithms on three datasets [47]: BA-SHAPES, TREE-CYCLES, and TREE-GRIDS. All the three datasets are synthetic datasets with ground truth which will help us to evaluate the quality of the explanations. Each dataset consists of a base graph, a particular type of motif attached to random nodes of the base graph and additional edges added between randomly chosen node pairs in the graph. These datsets are used in the node classification task, i.e., to predict whether a node belongs to that pre-defined motif or not. The datasets are described in the Appendix (Sec. A.2).

| Budget | Baseline | Fidelity | Time Taken (s) |
|---|---|---|---|
| | Random | 0.59 | 3.835 |
| | TopK | 0.58 | 11.73 |
| | Greedy | 0.54 | 27.34 |
| 3 | Shapley | 0.22 | 789.31 |
| | Banzhaf ($b$=0) | 0.24 | 224.44 |
| | Banzhaf ($b$=0.01) | 0.25 | 204.63 |
| | Banzhaf ($b$=0.05) | 0.25 | 184.43 |
| | Random | 0.66 | 3.776 |
| | TopK | 0.65 | 11.69 |
| | Greedy | 0.65 | 28.25 |
| 4 | Shapley | 0.28 | 795.04 |
| | Banzhaf ($b$=0) | 0.23 | 451.72 |
| | Banzhaf ($b$=0.01) | 0.24 | 443.70 |
| | Banzhaf ($b$=0.05) | 0.25 | 387.76 |
| | Random | 0.67 | 3.841 |
| | TopK | 0.66 | 11.66 |
| | Greedy | 0.66 | 28.34 |
| 5 | Shapley | 0.31 | 778.41 |
| | Banzhaf ($b$=0) | 0.32 | 746.10 |
| | Banzhaf ($b$=0.01) | 0.33 | 702.37 |
| | Banzhaf ($b$=0.05) | 0.33 | 639.52 |

Table 1: Results on *Fidelity (lower is better)* and *Running Time (lower is better)* for different budgets in TREE-GRID. For our method (Banzhaf), the results are shown with different values of thresholds.

| Budget | Baseline | Fidelity | Time Taken (s) |
|---|---|---|---|
| | Random | 0.46 | 2.36 |
| | TopK | 0.46 | 6.47 |
| | Greedy | 0.42 | 13.34 |
| 3 | Shapley | 0.38 | 187.75 |
| | Banzhaf ($b$=0) | 0.28 | 46.27 |
| | Banzhaf ($b$=0.01) | 0.30 | 44.09 |
| | Banzhaf ($b$=0.05) | 0.29 | 39.64 |
| | Random | 0.45 | 2.49 |
| | TopK | 0.44 | 6.64 |
| | Greedy | 0.42 | 13.70 |
| 4 | Shapley | 0.37 | 185.98 |
| | Banzhaf ($b$=0) | 0.39 | 68.43 |
| | Banzhaf ($b$=0.01) | 0.40 | 66.16 |
| | Banzhaf ($b$=0.05) | 0.39 | 60.59 |
| | Random | 0.45 | 2.37 |
| | TopK | 0.44 | 6.51 |
| | Greedy | 0.42 | 13.85 |
| 5 | Shapley | 0.41 | 183.40 |
| | Banzhaf ($b$=0) | 0.38 | 83.51 |
| | Banzhaf ($b$=0.01) | 0.39 | 80.47 |
| | Banzhaf ($b$=0.05) | 0.37 | 71.99 |

Table 2: Results on *Fidelity (lower is better)* and *Running Time (lower is better)* for different budgets in TREE-CYCLES. For our method (Banzhaf), the results are shown with different values of thresholds.

*5.1.2 Base Model.* As the base models to be explained, we train a 3-layer GCN model for BA-SHAPES and a 2-layer model for TREE-CYCLES and TREE-GRIDS. The different number of layers lead to better accuracy in the specific settings. Subsequently, we explain the models that are more accurate.

*5.1.3 Baselines.* We compare our method of Banzhaf values (*denoted as Banzhaf from now onwards*) with thresholds against four baselines. Note that our method is non-neural and does not involve any form of training for itself. Thus, we choose to compare against non-neural baselines to have a fair comparison. All the baselines algorithms produce a solution set with $k$ edges as the final explanation edges and they are as follows:
- **Random**: It selects the $k$ edges randomly from the graph.
- **TopK**: This approach selects the $k$ edges with highest utility values and returns them.
- **Greedy**: This approach selects the best edge based on the utility value iteratively at each step and adds it to the set of explanation edges, simultaneously removing it from the graph. The algorithm runs for $k$ steps to select $k$ edges.
- **Shapley**: For the shapley value, we use the Monte Carlo estimator as mentioned in the Sec. 4. In particular, to have an efficient method, we follow the procedure in [29].

*5.1.4 Evaluation Metrics.* We evaluate our algorithms on Fidelity [25], which is the proportion of nodes whose predicted class remains the same after the edges in the explanation set is deleted. Since we are generating counterfactual explanations, *a lower value of Fidelity*

is better. We also compare the running times of our algorithm and the baselines (in seconds).

*5.1.5 Parameters:* It is important to note that Shapley Value and Banzhaf Value are not learning algorithms, and therefore they do not have hyperparameters. Instead they have parameters that are not learnable. We have the following parameters:

**Budget ($k$):** All the algorithms take the budget as an input. The set of explanation edges returned by all the algorithms has size as the budget. This also can be seen as the explanation size.

**Threshold ($b$):** Following our theoretical results in Sec. ??, we vary the threshold parameter values in our experiments. The threshold values are based on the utility values of a coalition. We choose the coalitions based on the threshold value. We compute the ratio between the utility of a coalition to the probability value of the predicted class for the node. If the threshold is 0.1, then the selected coalitions have this mentioned ratio $\geq 0.1$.

**Other settings.** We describe other settings such as the coalition size, the number of coalitions, and how to choose the candidate set of edges in the Appendix (Sec. A.3). In all the experiments, the number of sampled coalitions sampled is 1500 and the coalition size for computing Banzhaf Values is equal to the budget unless specified otherwise (please see Sec. A.3 in the Appendix for more details). We also vary the coalition size and the number of sampled coalitions in the experiments (Sec. 5.4). To compute the performance measures, we randomly sample 50% of nodes of a given graph three times. We report the average of the results over these runs. We compute the

Shapley value of an edge as the average marginal contribution over the sampled permutations (we set it to 50).

## 5.2 Efficacy & Efficiency

We compare our proposed method using Banzhaf values with thresholding against the baselines using Fidelity. We also compute the running time of all the algorithms to illustrate their efficiency. Tables 1, 2, and 7 show the results for TREE-GRID, TREE-CYCLES, BA-SHAPES respectively (please see the Appendix for the results on BA-SHAPES).

We make a few interesting observations: **(1)** Our method based on Banzhaf values outperforms the other baselines in almost all cases as it captures the combinatorial effect of the deletions to produce a better counterfactual set. Random, TopK, and Greedy do not take into account this combinatorial effect. While Shapley takes into account this effect to some extent, it is an inferior method as can been seen from our theoretical results (Sec. 4). **(2)** Our method, Banzhaf consistently takes less running time than Shapley. This difference is further amplified with thresholding and we gain up to 10 times more efficiency. This is also consistent with our theoretical results where we show Banzhaf uses a lower number of samples.

## 5.3 Banzhaf vs Shapley: With Random Noise

In this section, we present the results of our method, Banzhaf and the best baseline Shapley with the presence of noise in the graph. We inject noise by adding edges between randomly sampled pairs of nodes in the graph and retrain the base GCN model. For this experiment, we add 5% of the total number of present edges randomly to the graph. Tables 3 and 4 present the results for TREE-CYCLES and BA-SHAPES respectively. We observe that, under noise, Banzhaf outperforms Shapley in most cases while being faster than Shapley. Similar results were observed for TREE-GRID dataset in Table 6 and 5. This is also consistent with our theoretical results in section 4.3 that Banzhaf with thresholding is robust with the presence of noise. In terms of efficiency, Banzhaf is much faster than Shapley. For instance, in the case of budget $k = 3$, Banzhaf with $b = .1$ is more than 30 times faster than Shapley.

## 5.4 Parameter variation

*5.4.1 Number of Coalitions.* A naive way to compute Banzhaf Values is to use all possible coalitions. However, this is not feasible as the number of possible coalitions is exponential in $|E|$, where $|E|$ is the number of candidate edges. Thus, we sample a fixed number of coalitions using the MSR principle as described in Sec. 4.1. Here, we vary the number of coalitions and evaluate the impact on Fidelity values. Figures 2a and 2b show Fidelity value as a function of number of coalitions for TREE-CYCLES and BA-SHAPES datasets respectively. We observe that variations in the coalition sizes produce similar results indicating they do not have a significant impact on the Fidelity value.

*5.4.2 Size of Coalitions.* Sampling the coalitions for calculating the Banzhaf value requires choosing the size of the coalition. We present the results of varying the coalition size as a function of the budget where coalition size = $k \pm 1$. Figures 3a and 3b show the Fidelity value as a function of the coalition size for the budget $k = 4$.

| Budget | Baseline | Fidelity Values | Time Taken (s) |
|---|---|---|---|
| 3 | Shapley | 0.37 | 310.51 |
| | Banzhaf ($b$=0) | 0.30 | 89.42 |
| | Banzhaf ($b$=0.01) | 0.31 | 83.90 |
| | Banzhaf ($b$=0.1) | 0.30 | 62.62 |
| 4 | Shapley | 0.36 | 299.98 |
| | Banzhaf ($b$=0) | 0.34 | 168.71 |
| | Banzhaf ($b$=0.01) | 0.33 | 162.73 |
| | Banzhaf ($b$=0.1) | 0.36 | 121.68 |
| 5 | Shapley | 0.36 | 300.46 |
| | Banzhaf ($b$=0) | 0.37 | 235.06 |
| | Banzhaf ($b$=0.01) | 0.36 | 233.07 |
| | Banzhaf ($b$=0.1) | 0.35 | 195.44 |

Table 3: With Noise (ratio = 5%): Fidelity and running time results in TreeCycles. Banzhaf outperforms Shapley in almost all cases while being faster. It shows the robustness of Banzhaf towards noise.

| Budget | Baseline | Fidelity Values | Time Taken (s) |
|---|---|---|---|
| 3 | Shapley | 0.43 | 627.02 |
| | Banzhaf ($b$=0) | 0.40 | 383.78 |
| | Banzhaf ($b$=0.01) | 0.40 | 104.83 |
| | Banzhaf ($b$=0.1) | 0.42 | 26.85 |
| 4 | Shapley | 0.39 | 645.55 |
| | Banzhaf ($b$=0) | 0.37 | 555.61 |
| | Banzhaf ($b$=0.01) | 0.37 | 283.83 |
| | Banzhaf ($b$=0.1) | 0.39 | 58.42 |
| 5 | Shapley | 0.40 | 636.01 |
| | Banzhaf ($b$=0) | 0.40 | 632.71 |
| | Banzhaf ($b$=0.01) | 0.40 | 419.11 |
| | Banzhaf ($b$=0.1) | 0.41 | 114.36 |

Table 4: With Noise (ratio = 5%): Fidelity and running time results in BA-SHAPES. Banzhaf outperforms Shapley in almost all cases while being faster. It shows the robustness of Banzhaf towards noise. For budget $k = 3$, Banzhaf with $b = .1$ is more than 30 times faster than Shapley.

Similar to Sec. 5.4.1, we observe that varying the size of coalitions does not produce a significant difference in the Fidelity value.

## 6 RELATED WORK

**Explainability in GNNs** While graph neural networks (GNNs) have gained significant attention recently due to their remarkable performance in various domains, such as natural language processing [43], their complex non-linear models make it challenging to understand the reasons behind their predictions. To address this, numerous explainability methods have been proposed; we refer the reader to a recent survey [16] for a detailed overview. Explainers for GNNs can be broadly classified into Factual and Counterfactual explainers. In our work, we focus on counterfactual explanations

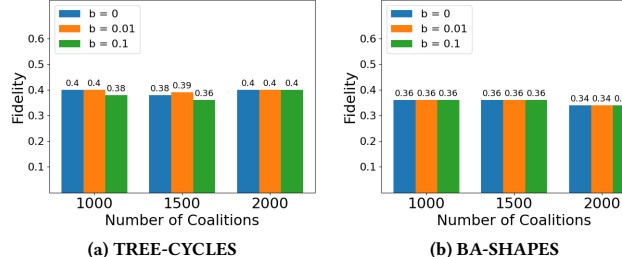

Figure 2: Results on Fidelity while varying the number of coalitions for three different thresholds in the (a) TREE-CYCLE and (b) BA-SHAPES datasets. We observe that the variation in the number of coalitions produce similar results for our Banzhaf method with budget ($k = 5$).

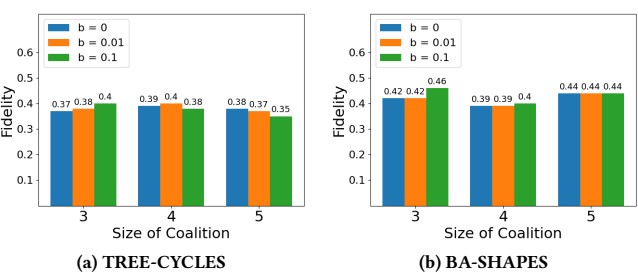

Figure 3: (a) Results on Fidelity while varying the size of coalition for three different thresholds in the (a) TREE-CYCLE and (b) BA-SHAPES datasets. We observe that slight variation in the size of coalition produce similar results for our Banzhaf method with budget ($k = 4$).

which have applications in areas such as drug discovery [25]. However, unlike most prior works which develop learning based methods [27, 47, 49], we focus on a non-learning based method which does not require training another graph.

**Factual Explainers for GNNs.** Factual explainers usually provide a subset of the input features as explanations. Unlike counterfactual ones, factual explanations seek to answer the reasoning question from the existing data: *what are the input features X that are responsible for the current output Y?* These features could be any substructure such as nodes, edges or subgraphs. For example, *gradient-based methods* such as guided-bp [3] and grad-CAM[34] measure how sensitive the output is to the input features. They use gradients to rank the input features. GraphSVX[9], ReLex[50], DnX[33] are *surrogate-based methods* and use a simpler surrogate model to explain the output. *Perturbation-based methods* compute explanations by introducing small changes in the input features to see how the output changes. These methods include GraphMask[35], GNNExplainer[47], and PGExplainer[27]. *Generation-based methods* [23, 24, 41] use generative methods to construct graphs that act as explanations. Our work is complementary to these methods since it aims to identify counterfactual explanations which are responsible for changing the current predictions.

**Counterfactual Explainers for GNNs** Counterfactual explanation aims to answer - *How should we change the input X to X'*

so that output changes from $Y$ to $Y'$. Existing counterfactual explainers of GNNs include CF-GNNExplainer [25], which generates minimal perturbations to the input graph data to change the prediction, and RCExplainer [2], which produces robust counterfactual explanations by modeling the common decision logic of GNNs on similar input graphs. *Search-based methods* such as MMACE[42] and MEG[30] search the counterfactual space of candidates and generate explanations. *Neural Network based* methods such as CLEAR[28] use neural networks to generate the counterfactual explanations with causality by identifying a subset of edges which when removed change the prediction. Global Counterfactual Explainer [14] addresses the limitations of instance-specific local reasoning by studying global counterfactual explainability of GNNs.

**Game-theoretic Methods in GNN Explanations.** Previous research has proposed multiple methods inspired by game theory to generate explanations that make use of semivalues. GraphSVX [9] is a post-hoc local model-agnostic explanation method specifically designed for GNNs. It uses Shapley values to capture the "fair" contribution of each feature and node towards the explained prediction. Another method, GStarX [49], leverages graph structure information to improve explanations by defining a scoring function based on a new structure-aware value from cooperative game theory. Both of these are factual explainers whereas we focus on building counterfactual explanation. Broadly speaking, **Semivalues** offer a versatile framework for assessing the contributions of individual agents in cooperative settings, and the choice of utility functions can be tailored to the specific context. For instance, in data valuation problems, the utility function often represents the quality or importance of a subset of data points, where the utility is determined by metrics like test accuracy, information gain, or relevance to a particular task [20, 20, 39]. In voting games, the utility function is typically associated with the voting power of players, quantified by their influence on the collective decision-making process [7, 32]. To the best of our knowledge, *our work is the first to employ Banzhaf values to generate counterfactual explanations for GNNs.*

# 7 CONCLUSIONS

Graph Neural Networks (GNNs) have proven to be a valuable tool for prediction tasks in complex networks. Nevertheless, their decision-making processes have remained somewhat black-box to the users, posing challenges in understanding their prediction outcomes. To address this interpretability issue, we have introduced a novel approach to build a counterfactual explainer using thresholded Banzhaf Values for the node classification task. While several methods have been proposed in the literature, they mostly depend on learning-based approaches that needs additional training, where as, our approach does not require training. We also have shown that our proposed method based on Banzhaf is faster than other game-theoretic measure such as Shapley value and more robust in the presence of noise despite the wide-spread usage of later. In practice, our method Banzhaf produces high-quality results either better or comparable to Shapley while being up to 10 times faster. As a future direction, it will be interesting to see if Banzhaf value-based method can be effective for building counterfactual explainer for other tasks.

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

# A APPENDIX

## A.1 Robustness Framework

As mentioned earlier, we use the robustness framework from [39] to show the effect of thresholding on semivalues. In particular, we use the following definitions.

**DEFINITION 1.** *The scaled difference between two edges $i$ and $j$ is*

$$D_{i,j}(U; w) := n(\phi(i; w) - \phi(j; w))$$

$$= \sum_{k=1}^{n-1}(w(k) + w(k+1))\binom{n-2}{k-1}\Delta_k^{i,j}(U),$$

*where* $\Delta_k^{i,j}(U) := \sum_{\substack{|S|=k-1 \\ S \subseteq N\setminus\{i,j\}}} [U(S \cup \{i\}) - U(S \cup \{j\})].$

$\Delta_k^{i,j}(U)$ represents the average distinguishability between $i$ and $j$ on size-$k$ sets using the noiseless utility function $U$. Let $\widehat{U}$ denote a noisy estimate of $U$.

**DEFINITION 2.** *We say a pair of edges $(i, j)$ is $\tau$-distinguishable by $U$ if and only if $\Delta_k^{i,j}(U) \geq \tau$ for all $k \in \{1, \ldots, n-1\}$.*

**DEFINITION 3.** *Given $\tau > 0$, we define the safety margin of a semivalue for a pair of edges $(i, j)$ as*

$$Safe_{i,j}(\tau; w) := \min_{U \in U(\tau)_{i,j}} \min_{\widehat{U} \in \{\widehat{U}: D_{i,j}(U;w) D_{i,j}(\widehat{U};w) \leq 0\}} ||\widehat{U} - U||. \tag{6}$$

*And the safety margin of a semivalue is defined as*

$$Safe(\tau; w) := \min_{i,j \in N, i \neq j} Safe_{i,j}(\tau; w). \tag{7}$$

Now, we are ready to show the proof of Theorem 2 that uses the definitions mentioned above. We begin by introducing some notations for our theoretical analysis. Let $\widehat{U}$, denote the noisy utility function and let $x = U - \widehat{U}$ denote the noise. We also assume the $B$ vector with the same dimension as $U$ and for all $b_i \in B$ $b_i = b$ where $b$ is the constant threshold. Now after applying the threshold, let $U' = max(U-B, 0)$ and $\widehat{U'} = max(\widehat{U}-B, 0)$. Finally, let $x' = U' - \widehat{U'}$.

**THEOREM 2.** *Adding the threshold to the utility function doesn't change the safety margin for any semivalue $w$.*

PROOF.

$$D_{i,j}(U; w) = a^T U$$

where each entry of $a$ corresponds to a subset $S \subseteq N$. We use $a[S]$ to denote the value of $a$'s entry corresponds to $S$. For all $S \subseteq N\setminus\{i,j\}, a[S \cup i] = w(|S| + 1) + w(|S| + 2)$ and $a[S \cup j] = -(w(|S| + 1) + w(|S| + 2))$, and for all other subsets $a[S] = 0$. Let matrix $A = aa^T$.

$$D_{i,j}(U'; w) D_{i,j}(\widehat{U'}; w) = (a^\tau U')(a^T \widehat{U'})$$

$$= (a^T U')^T (a^T \widehat{U'})$$

$$= U'^T aa^T \widehat{U'}$$

$$= U'^T A \widehat{U'}$$

$$= U'^T A(U' - x')$$

Now, from the definition of x' and x, we can see that $||x'|| = ||x||$ for the case where $\forall u \in U, u > b$. Further if $\exists u \in U$ where $u < b$ or $\exists \hat{u} \in \widehat{U}$ where $\hat{u} < b$ then $||x'|| < ||x||$. This means adding the threshold to the utility function leads to $||x'|| \leq ||x||$. Combining the above analysis of threshold with the proof of lemma C.1 from [39], -

$$||x|| \geq ||x'|| \geq \sqrt{\frac{\left|\widehat{U'}^T A U'\right|}{a^T a}} \tag{8}$$

Since the safety margin considers minimum required perturbation value of $x'$, we can consider the case where $x' = x$ (i.e thresholds have no effect on $\widehat{U} - U$ and $||x'|| = ||x||$).

$$Safe(\tau; w) = \tau \sqrt{\frac{\left(\sum_{k=1}^{n-1}\binom{n-2}{k-1}(w(k) + w(k+1))\right)^2}{\sum_{k=1}^{n-1}\binom{n-2}{k-1}(w(k) + w(k+1))^2}} \tag{9}$$

for any $\tau > 0$ □

## A.2 Description of Datasets

**BA-SHAPES:** This consists of a base Barabasi-Albert (BA) graph consisting of 300 nodes. The motif is a house shaped motif consisting of five nodes. There are three types of nodes in the house motif corresponding to their position: one node is at the top, two nodes in the middle and the remaining two nodes at the bottom. The graph has 80 motifs and the nodes belong to four different classes including presence in the base graph and other 3 classes are for the three types of nodes in the house motif.

**TREE-CYCLES:** This consists of a base balanced binary tree graph consisting of 511 nodes. The motif is a cycle containing 6 nodes and there are 60 motifs in the graph. The nodes in the dataset belong to 2 classes: nodes in the base graph are assigned class 0 and nodes in the motif are assigned class 1.

**TREE-GRID**: The base graph is same as in TREE-CYCLES and the nodes are in two different classes. The motif is a 3x3 grid containing nine nodes and there are 80 motifs in the graph.

## A.3 Other Settings

Here, we elaborate on additional details of other parameters in Banzhaf and Shapley:

**Coalition size:** As stated in Sec. 5.4.2,when calculating the Banzhaf value, we need to sample coalitions of edges. The number of edges in a sampled coalition is given by the Coalition size parameter.

**Number of Coalitions:** From Sec. 5.4, we infer that we need to sample a fixed number of coalitions to calculate the Banzhaf value to avoid running into exponential running time. This number is given by the Number of Coalitions parameter.

**Hop Size:** This parameter controls the no of candidate explanation edges. In particular, we find the induced subgraph of the node under consideration upto a fixed number of hops. All the edges in the induced subgraph are the candidate explanation edges.

| Budget | Threshold | Fidelity | Total Time Taken |
|---|---|---|---|
| | Shapley | 0.37 | 787.86 |
| 3 | Banzhaf ($b$=0.) | 0.31 | 219.25 |
| | Banzhaf ($b$=0.01) | 0.30 | 233.92 |
| | Banzhaf ($b$=0.1) | 0.29 | 88.64 |
| | Shapley | 0.35 | 852.82 |
| 4 | Banzhaf ($b$=0.) | 0.30 | 276.38 |
| | Banzhaf ($b$=0.01) | 0.30 | 202.28 |
| | Banzhaf ($b$=0.1) | 0.29 | 109.28 |
| | Shapley | 0.30 | 891.91 |
| 5 | Banzhaf ($b$=0) | 0.25 | 330.86 |
| | Banzhaf ($b$=0.01) | 0.24 | 258.99 |
| | Banzhaf ($b$=0.1) | 0.23 | 127.16 |

**Table 6: With Noise (ratio = 5%): Fidelity and running time results in TREE-GRIDS. Banzhaf outperforms Shapley in almost all cases while being faster. It shows the robustness of Banzhaf towards noise. For budget $k = 3$, Banzhaf with $b = .1$ is more than 10 times faster than Shapley.**

| Budget | Baseline | Fidelity | Time Taken (s) |
|---|---|---|---|
| | Random | 0.53 | 2.87 |
| | TopK | 0.51 | 20.38 |
| | Greedy | 0.36 | 56.53 |
| 3 | Shapley | 0.46 | 520.27 |
| | Banzhaf ($b$=0) | 0.35 | 309.88 |
| | Banzhaf ($b$=0.01) | 0.35 | 108.75 |
| | Banzhaf ($b$=0.05) | 0.35 | 35.82 |
| | Random | 0.58 | 2.85 |
| | TopK | 0.59 | 20.40 |
| | Greedy | 0.40 | 69.85 |
| 4 | Shapley | 0.39 | 506.85 |
| | Banzhaf ($b$=0) | 0.39 | 469.24 |
| | Banzhaf ($b$=0.01) | 0.39 | 290.24 |
| | Banzhaf ($b$=0.05) | 0.39 | 97.19 |
| | Random | 0.55 | 2.84 |
| | TopK | 0.58 | 20.67 |
| | Greedy | 0.35 | 80.57 |
| 5 | Shapley | 0.35 | 552.31 |
| | Banzhaf ($b$=0) | 0.36 | 506.04 |
| | Banzhaf ($b$=0.01) | 0.36 | 381.31 |
| | Banzhaf ($b$=0.05) | 0.35 | 251.87 |

**Table 7: Results on *Fidelity (lower is better)* and *Running Time (lower is better)* for different budgets in BA-SHAPES. For our method (Banzhaf), the results are shown with different values of thresholds.**

## A.4 Additional Experimental Results

In this section, we present additional results of efficacy(Sec.5.2) and random noise(Sec. 5.3).

**Efficacy and efficiency.** Table 7 presents the fidelity and time values of running our algorithm Banzhaf with threshold against other baseline algorithms(Sec. 5.1.3) on the BA-SHAPES dataset. Consistent with our observations in Section 5.2, we observe that Banzhaf performs better than all the baselines in almost all the cases while consistently having a much lower running time than Shapley value.

**With random noise.** Tables 5 and 6 show the results of adding noise on the TREE-GRID dataset for noise ratios 10% and 5% respectively. Again, consistent with our observations in Section 5.3, we observe that Banzhaf consistently gives better fidelity value than Shapley while having a much lower running time. This proves that Banzhaf is robust in the presence of noise.

| Budget | Threshold | Fidelity | Total Time Taken |
|---|---|---|---|
| | Shapley | 0.41 | 632.13 |
| 3 | Banzhaf ($b$=0) | 0.32 | 77.91 |
| | Banzhaf ($b$=0.01) | 0.29 | 57.81 |
| | Banzhaf ($b$=0.1) | 0.28 | 34.96 |
| | Shapley | 0.42 | 657.28 |
| 4 | Banzhaf ($b$=0) | 0.40 | 114.43 |
| | Banzhaf ($b$=0.01) | 0.37 | 71.41 |
| | Banzhaf ($b$=0.1) | 0.35 | 38.94 |
| | Shapley | 0.41 | 680.41 |
| 5 | Banzhaf ($b$=0) | 0.38 | 128.03 |
| | Banzhaf ($b$=0.01) | 0.34 | 90.11 |
| | Banzhaf ($b$=0.1) | 0.32 | 44.46 |

**Table 5: With Noise (ratio = 10%): Fidelity and running time results in TREE-GRIDS. Banzhaf outperforms Shapley in almost all cases while being faster. It shows the robustness of Banzhaf towards noise. For budget $k = 3$, Banzhaf with $b = .1$ is more than 10 times faster than Shapley.**

