# OpenReview forum: "Game-theoretic Counterfactual Explanation for Graph Neural Networks"
_ACM.org/TheWebConf/2024/Conference — TheWebConf24_

### Official Review · Reviewer_Pn9d · 2023-11-19

**Novelty:** 6
**Technical Quality:** 5

**Review:**

In this paper, the authors focus on the interpretability of GNNs, which is a very important research topic. Unlike previous research works, the authors propose a novel approach to build a counterfactual explainer using thresholded Banzhaf Values for the node classification task. They illustrate the proposed approach through rigorous theoretical proofs. In addition, the authors significantly outperform other baselines in terms of time efficiency.

**Questions:**

1. The authors compare fewer baselines, why don't you compare more last baselines? Is this field less popular?

2. The authors performed experiments on only 3 artificial datasets. Why don't you conduct experiments on real-world datasets? The ultimate goal of research on the interpretability of graph neural networks is to enhance human trust in the model in real-world applications. The lack of experiments on real-world datasets makes it difficult for me to believe that the authors' method can work in the real world.

3. Can the authors' proposed method be generalized to graph-level tasks?

4. Why is b set to 0.05 in Tables 1 and 2, but 0.1 in Tables 3 and 4?

5. Line 679 has a citation error.

**Reviewer Confidence:**

3: The reviewer is confident but not certain that the evaluation is correct

**Scope:**

4: The work is relevant to the Web and to the track, and is of broad interest to the community

---

### Official Review · Reviewer_iPhy · 2023-11-21

**Novelty:** 4
**Technical Quality:** 5

**Review:**

This paper addresses the counterfactual explanation of node classification using GNN model. Instead of using training-based methods, it uses semi-values and is hence training-free and efficient. The proposed method, "Thresholded Banzhaf Values", is shown to be sample efficient theoretically and demonstrated to be so experimentally, in particular against the Shapley value, which is the previous attempts to solve the problem of counterfactual explanability on GNNs. In addition, the paper attempts to show the superiority of the Banzhaf value over the Shapley value by using experiments on explanability benchmarks and then in the presence of random noise. While the Banzhaf value seem to outperform the Shapley value under some budgets, it seems that the performance gain is not significant. While the theoretical part of this paper is rather solid, some questions remain about the specifics of experimental designs and the interpretation of the experimental results.

**Questions:**

1. How many repeated experiments are performed for each reported results, and how is the performance' stability/variance? The estimation of the values can be stochastic, so it is important to report the performance averaged over multiple runs.

2. It seems that the main contribution should be the efficiency of using the Banzhaf value over the Shapley value, as the performance gain is not significant, is that the case?

3. The methods such as Random, Top-K, Greedy, while poor in terms of fidelity, is much faster to run. So if the concern is efficiency, it is worthwhile to design a new metrics that consider both fidelity and "time taken" together.  It is important to show the model's superiority over others in this 'holistic' way.

**Ethics Review Description:**

No Ethics Concerns

**Reviewer Confidence:**

3: The reviewer is confident but not certain that the evaluation is correct

**Scope:**

3: The work is somewhat relevant to the Web and to the track, and is of narrow interest to a sub-community

---

### Official Review · Reviewer_5nmi · 2023-11-22

**Novelty:** 4
**Technical Quality:** 6

**Review:**

Contributions:

The authors propose a semivalue-based method for generating counterfactual explanations (CFE) for node classification tasks that does not require additional learning. They design a thresholding method for computing Banzhaf values, and they find that computing Banzhaf values (rather than Shapley values) requires a lower sample complexity and may be more efficient (up to 4 times faster) and robust.

My recommendation is based on S1, S2, W1, W2, W3. I am happy to raise my scores based on the authors' responses to my questions and clarification/justification of W1, W2, W3.

Quality:

Pros:
- (S2) The theoretical analysis of computational efficiency generally looks correct (Section 4.1).

- The authors theoretically show that using thresholding to compute Banzhaf values does not change their safety margin.

- (S1) Banzhaf vallues are empirically more efficient.

Cons:
- Lines 198-205: The authors state that a semivalue maps subsets of players to a real number, but a semivalue should take a characteristic function/game as input.

- (W1) Experiments: The authors should also include other edge explanation methods that are not based on Shapley values (even if they are not counterfactual explanations) as baselines (e.g., GStarX). The authors could also run more experiments to see how sensitive Banzhaf values are to the budget, as it appears from Tables 1 and 2 that Banzhaf values vary significantly between a budget of 3 and 5. Why do the authors not report the standard error over the three samples (lines 693-694)? The robustness results for Banzhaf values in Tables 3 and 4 are not convincingly lower than Shapley values without reporting the standard error over the samples.

Clarity:

Cons:
- The properties of semivalues (Section 2.2.1) are poorly stated by the authors. (1) should be $\Phi_i$ rather than $\Phi$. The symbol for the characteristic function changes from $v_1$ to $U$ to $v$. Please provide the domain and co-domain of $U$ and a high-level description of what it represents.

- Line 399: Given that the superiority of Banzhaf values in terms of efficiency hinges on MSR being numerically unstable for Shapley values, the authors should elaborate further on this numerical instability and why it is not a problem for Banzhaf values.

- The authors leverage numerous results from [1] in their theoretical analysis and should thus clearly restate these results in the paper before using them.

Originality:

Pros:
- The authors contribute a thresholding method for efficiently and robustly computing Banzhaf values.

Cons:
- (W2) The authors adopt the MSR estimator and much theoretical analysis for Banzhaf values from [1].

Significance:

Pros:
- Current learning-based approaches for CFE often require additional training and may not be interpretable themselves.

- Banzhaf values are more intuitive than Shapley values for edge attribution because node classification should be invariant to the ordering of edges.

Cons:
- (W3) The authors should discuss the limitation and societal implications of their work.

[1] Jiachen T Wang and Ruoxi Jia. 2023. Data banzhaf: A robust data valuation framework for machine learning. In International Conference on Artificial Intelligence and Statistics. PMLR, 6388–6421.

[2] Zhang, Shichang, et al. "Gstarx: Explaining graph neural networks with structure-aware cooperative games." Advances in Neural Information Processing Systems 35 (2022): 19810-19823.

EDIT: I have read the authors' rebuttal.

**Questions:**

Please see Review (above).

**Reviewer Confidence:**

2: The reviewer is willing to defend the evaluation, but it is likely that the reviewer did not understand parts of the paper

**Scope:**

3: The work is somewhat relevant to the Web and to the track, and is of narrow interest to a sub-community

---

### Official Review · Reviewer_ZbQS · 2023-11-26

**Novelty:** 4
**Technical Quality:** 4

**Review:**

This paper proposes to use Banzhaf value to find top-k important edges in GNN predictions. The proposed method approximate Banzhaf value using MSR estimator and incorporates a threshold to the utility function. Experiments show the proposed method has lower running time than the MC estimator for Shapley value and better CFE performance than methods that do not need training.


Strength

- This work proposes a method that enables us to obtain counterfactual examples for graph data without extra training.

- This work shows theoretical results on the number of calls of the utility function needed to obtain the top-k important edges.

Weakness

- this work only considers counterfactual examples that are obtained from deleting edges, ignoring other possible counterfactual examples which can only be obtained by modifying features and adding edges.

- the problem is defined as finding top-k important edges that contribute the most to the prediction of class c. However, this does not really align with the definition of CFE in the literature, which means with top-k edges removed, it is not necessary to have the predicted label flipped too. I would suggest the authors to replace the use of the term counterfactual explanation to something else.

- In the experiment, even if the method does not need training, it will be better to compare with some of the representative methods that need training as an upperbound of performance.

- The error of MSR estimator in approximating the true Banzhaf value is not discussed.

**Questions:**

- The notation is quite confusing. In L195-196 v is a function, but v is a node in L254 etc.

- L322-L331 It would be better to elaborate the difference between Banzhaf value and Shapley value formally with solid notations. The text description is very hard to comprehend.

- What is the full name of MSR estimator?

- Notation mismatch in L391 and L393.

- L404, I am confused. Is \phi_{MC} for Shapley value or Banzhaf value? As L371 mentioned it is for Banzhaf value.

- L425, what is \beta? It is not defined.

- L679 there is a question mark.

- In pratice, what is the best way to select a reasonable threshold?

- The time taken of Banzhaf increases with budget while Shapley does not in Table 1-4. Does this imply that, when budget is large enough, Banzhaf can be more expensive than Shapley?

- In the MSR estimator, I guess the expectation is taken over S, where S is the set of subgraphs that have at least one node as the neighbor of i. It would be great for the authors to make this clear.

**Reviewer Confidence:**

4: The reviewer is certain that the evaluation is correct and very familiar with the relevant literature

**Scope:**

4: The work is relevant to the Web and to the track, and is of broad interest to the community

---

### Official Review · Reviewer_NBys · 2023-11-28

**Novelty:** 5
**Technical Quality:** 5

**Review:**

Strongness:
1. It provides a comprehensive comparison of the proposed method with existing baselines like Shapley values, demonstrating its efficiency and effectiveness.
2. The application of Banzhaf values for generating counterfactual explanations in GNNs is a novel approach.
3. The paper introduces new insights into the efficiency and robustness of Banzhaf values compared to Shapley values in the context of GNNs.
4. The study addresses an important aspect of AI interpretability, contributing to the understanding of GNN predictions, which is crucial for their adoption in critical applications.
5. The proposed method could be particularly useful in domains where understanding the rationale behind AI decisions is as important as the decisions themselves.

Weakness:
1. The technical nature and the complexity of the game-theoretic concepts might make the paper challenging for readers not familiar with these areas.
2. The foundational concepts of GNNs and counterfactual explanations are established areas, so the paper's contribution lies in the novel application of these concepts rather than in the creation of new theories.
3. The proposed method could be particularly useful in domains where understanding the rationale behind AI decisions is as important as the decisions themselves.

**Questions:**

1. How well does your method perform on real-world datasets, considering their complexity and variability compared to synthetic datasets?
2. How scalable is your method in terms of graph size and complexity? Are there any limitations when it comes to larger or more complex networks?
3. Beyond the Shapley value, did you compare the Banzhaf value approach with other explainability methods? How does it fare against these methods?
4. How does the structure of the graph (e.g., density, connectivity) affect the counterfactual explanations generated by your method?
5. How robust is your method in the presence of noisy or incomplete data, which is common in real-world scenarios?
6. How interpretable are the explanations generated by your method for stakeholders or decision-makers who may not be experts in graph theory or machine learning?
7. Have you investigated the potential for biases in the explanations generated by your method? How can such biases be identified and mitigated?

**Reviewer Confidence:**

3: The reviewer is confident but not certain that the evaluation is correct

**Scope:**

3: The work is somewhat relevant to the Web and to the track, and is of narrow interest to a sub-community

---

### Decision · Program_Chairs · 2024-01-22

**Decision:**

Accept

**Comment:**

The paper presents a new approach to counter factual analysis for GNNs via the use of Banzhaf values. The authors show the superiority of this approach as compared to the standard approach of computing Shapely values. All the reviewers agreed that this is a novel technical contribution. The authors are recommended to take into account the readability related suggestions from the reviewers.